# SRC-Family Kinases in Acute Myeloid Leukaemia and Mastocytosis

**DOI:** 10.3390/cancers12071996

**Published:** 2020-07-21

**Authors:** Edwige Voisset, Fabienne Brenet, Sophie Lopez, Paulo de Sepulveda

**Affiliations:** INSERM U1068, CNRS UMR7258, Aix-Marseille Université UM105, Institute Paoli-Calmettes, CRCM—Cancer Research Center of Marseille, U1068 Marseille, France; edwige.voisset@inserm.fr (E.V.); fabienne.brenet@inserm.fr (F.B.); sophie.lopez@inserm.fr (S.L.)

**Keywords:** protein kinase, SFK, FLT3, KIT, AML, mastocytosis, kinase inhibitor

## Abstract

Protein tyrosine kinases have been recognized as important actors of cell transformation and cancer progression, since their discovery as products of viral oncogenes. SRC-family kinases (SFKs) play crucial roles in normal hematopoiesis. Not surprisingly, they are hyperactivated and are essential for membrane receptor downstream signaling in hematological malignancies such as acute myeloid leukemia (AML) and mastocytosis. The precise roles of SFKs are difficult to delineate due to the number of substrates, the functional redundancy among members, and the use of tools that are not selective. Yet, a large num ber of studies have accumulated evidence to support that SFKs are rational therapeutic targets in AML and mastocytosis. These two pathologies are regulated by two related receptor tyrosine kinases, which are well known in the field of hematology: FLT3 and KIT. FLT3 is one of the most frequently mutated genes in AML, while KIT oncogenic mutations occur in 80–90% of mastocytosis. Studies on oncogenic FLT3 and KIT signaling have shed light on specific roles for members of the SFK family. This review highlights the central roles of SFKs in AML and mastocytosis, and their interconnection with FLT3 and KIT oncoproteins.

## 1. Introduction

The SRC-family of kinases (SFKs) comprises eight members: BLK, FGR, FYN, HCK, LCK, LYN, SRC and YES. They are further divided into two subfamilies: the SRC-related subfamily, also called Src-A (FGR, FYN, SRC and YES), and the LYN-related subfamily, also called Src-B (BLK, HCK, LCK and LYN). While SRC, FYN and YES are ubiquitously expressed in mammals, the other members of the family show tissue-specific features. LYN and FGR are mainly expressed in the hematopoietic system [1,2], while HCK is preferentially expressed in myeloid lineages [3,4], BLK in B cells [5], and LCK in T cells [6]. All family members share similar structural features, with a unique NH2-terminal region and three well characterized protein domains: two protein interaction modules, SH3 and SH2, and the tyrosine kinase catalytic domain SH1 [7]. SFKs are covalently modified following translation, with the addition of a lipid moiety (myristoylation and/or palmitoylation at the NH2-terminus), allowing these signaling molecules to be anchored to the inner face of cell membranes. They relay signals from cell surface receptors through a multitude of interactors and substrates. SFKs have pleiotropic functions in cells. Their contribution to cell transformation has been known for 40 years, since the discovery of v-SRC, the oncogene product of the Rous sarcoma virus.

Deciphering the role of SFKs in normal physiology, and in cancer, is highly complex. This is due to several factors, including functional redundancy between related members, inter-regulation between members, the complexity of the substrate network in SFKs, and their pattern of expression [8]. Also, because of the limited number of tools available, SFKs have sometimes been studied as a group, without distinguishing between the different family members. The protein domains of SFKs are sufficiently similar for many of them to interact with the same ligands. They also have numerous substrates, some of which are common to several members [9]. Indeed, SFKs phosphorylate many proteins, ranging from enzymes to adaptors and structural proteins. SFKs regulate many signaling pathways by direct activation of proteins (e.g., STAT proteins or Rho family regulators), or through the modification of scaffold/adaptor proteins that build up signalosomes responsible for the initiation of multiple signaling cascades. Furthermore, analyses of mouse knock-out models have revealed both unique features and redundancy among SFKs. Finally, several generations of SFKs inhibitors originally thought to be highly selective have been shown to be promiscuous, and while they have shed light on some aspects of SFK functions, they have also led to confusion in the field, due to the possibility of misinterpretation. In addition to the use of mouse models, novel reagents and experimental approaches targeting single members of the family are starting to clarify the contribution of individual SFKs, allowing the investigation of the distinct contributions of each member of the family.

SFKs switch from active forms (phosphorylated at tyrosine residue 416 in SRC, referred to as pY416, and homologous residues of the other members of the family) to inactive forms (phosphorylated at the C-terminus residue Y527) [8]. The transition between states is dynamic: it is regulated through interactions with upstream regulators which release the structural constraints imposed by the SH2 or SH3 domains on the catalytic domain, and by the proximity of negative regulators such as CSK kinase or tyrosine phosphatases. The activation status of SFKs can be monitored using specific antibodies that allow detection of the active pY416 forms and the inactive form pY527. Other tools commonly used in studies on the role of SFKs include a panel of tyrosine kinase inhibitors, dominant-negative forms of SFKs, and RNA interference approaches which allow specific targeting of individual SFK.

Aberrant activation of SFKs has been observed in various types of cancer, thus affecting their crucial roles in proliferation, survival, adhesion, chemotaxis, invasion and metastasis. It should be noted that, to date, no mutation of SFK genes has ever been identified in AML [10,11,12]. Only an ETV6-LYN gene fusion has been reported, in primary myelofibrosis [13] and in acute myeloid leukemia [14].

In the present review, we will summarize the current knowledge of SFK functions in acute myeloid leukemia and mastocytosis, with a focus on oncogenic FLT3 and KIT receptor signaling.

## 2. SFKs in Acute Myeloid Leukemia (AML)

Acute myeloid leukemia is characterized by the accumulation of immature hematopoietic cells in the bone marrow and blood. The disease is complex, with prognosis defined by various molecular mutations and cytogenetic profiles. The molecular heterogeneity of the disease, as well as clonal heterogeneity within patients, is a challenge when developing therapeutic strategies. Overall, AML remains a deadly disease, due to frequent relapse, and to the emergence of clones resistant to standard chemotherapy. Driver mutations in the FLT3 receptor occur in 25–30% of patients [15]. The most frequent type of mutation, designated FLT3-ITD (internal tandem duplication), is correlated with a frequent rate of relapse and a poor prognosis [16]. The in-frame duplication of amino-acids occurs in the juxtamembrane region of the receptor, a domain which is required to maintain the catalytic domain in an inactive conformation in the absence of the ligand. As a consequence of the ITD mutation, FLT3 is activated and downstream signaling occurs independently of ligand binding [17]. In addition to pathways activated by the wild-type receptor, FLT3-ITD oncoprotein activates illegitimate pathways, including STAT5, and two other non-receptor tyrosine kinases, FES [18] and SYK [19]. This aberrant activation of FLT3, in cooperation with other mutations, is responsible for increased survival and proliferation of blast cells.

Point mutations at codon D835, which mainly result in D to V amino-acid change, constitute another hotspot of FLT3 mutation. These account for 5% of AML cases, and, unlike ITD mutations, do not exhibit a clear influence on prognosis. Small chemical inhibitors of FLT3 (e.g., quizartinib, gilteritinib) are efficient reducers of the leukemic load in patients; however, during the following months rapid relapse is inevitable, due to the selection of cells that have developed resistance, or the selection of minor clones with no FLT3 mutations.

Receptor tyrosine kinases such as FLT3 recruit intracellular signaling mediators directly, through binding to phosphorylated tyrosine residues which act as docking sites for SH2-containing proteins, or indirectly, through the formation of large protein complexes around adaptor/scaffold proteins, which constitute efficient signaling hubs. FYN, HCK, LCK, LYN, FGR and SRC interact, through their SH2 domains, with the docking sites Y589, Y591 and Y599, which are located in the juxtamembrane region of FLT3 [20,21]. FLT3 activates SFKs, either by wild-type (WT) receptor upon ligand-stimulation [22], or via the FLT3-ITD oncoprotein [23] (Figure 1).

KIT is another transmembrane receptor tyrosine kinase (RTK), structurally related to FLT3, CSF1R and PDGFR, with pleiotropic cellular functions, including cell differentiation, migration, survival and proliferation [24]. In AML, mutations of this RTK are predominantly found with either t(8;21) or inv(16) chromosomal rearrangements, leading to the fusion proteins RUNX1-RUNX1T1 and CBFB-MYH11, respectively, referred to as core binding factor (CBF) AML. KIT mutations in CBF-AML have a frequency of 30% to 40%, and are predominantly located in the extracellular domain (exon 8) and the tyrosine kinase domain (exon 17 mutations, resulting in either D816V or N822K substitutions, for example). At the protein level, they are responsible for the constitutive activation of KIT even in the absence of ligand. The presence of KIT mutations in CBF-AML is associated with a poorer prognosis [15].

### 2.1. Expression of SFKs in AML

Several SFKs are expressed in AML. Previous studies had suggested that expression of FGR and HCK in acute myeloid leukemia blasts was associated with early commitment and differentiation events in the monocytic and granulocytic lineages [25]. Multiple studies have now shown that FGR, FYN, LYN and HCK are the most widely expressed SFKs in myeloid cells and AML, as monitored by RNA levels [25,26,27]. At the protein level, LYN is the most consistently expressed at high levels, but HCK and FGR are also strongly expressed in a significant proportion of AML patients [28,29]. In addition, SRC and LCK can also be expressed in AML specimen and cell lines. For instance, the widely used MV4-11 cell line expresses significant levels of SRC protein in addition to FGR, HCK, LYN and FYN [30,31].

Interestingly, analyses of gene expression in patient cohorts have produced evidences for the involvement of SFKs in AML. For instance, HCK is part of an LSC-enriched signature in AML [32]. Furthermore, mRNA expression of myeloid SRC-family kinases HCK, FGR and LYN correlated with patient survival, which suggests that these three members are prognostic factors and that they play a functionally important role in disease progression [27].

### 2.2. SFKs Are Activated in AML

SFKs are not only expressed, but also activated, in most AML primary samples and cell lines regardless of their cytogenetic, molecular or cytological background [28,29,33]. LYN is frequently expressed and activated in AML samples [22,28]. Furthermore, the reduction of LYN expression, using RNA interference either in AML cell lines or primary specimens, has shown that LYN is a major player in this disease, relaying signaling towards mTOR and cell proliferation [28,34,35]. It must be emphasized that, due to the variety of mutational profiles they present, AML samples are quite heterogeneous, and thus each sample or cell line may have its own specific pattern of SFK expression and activation.

Considering the importance of SFK signaling in solid cancers [36,37], the hyperactivation of SFKs in AML suggests an active contribution to the leukemic phenotype. Indeed, SFKs seem to play essential roles in several oncogenic pathways, and in the maintenance of the leukemic phenotype in AML. This is supported by evidence based on RNA interference data, obtained using various cell lines, as indicated above for LYN: for instance, in MV4-11 cells, FGR and HCK are required for cell proliferation [30,31]. Remarkably, reduction of FGR, HCK or LYN expression in primary AML samples resulted in increased apoptosis, reduced growth, and impaired colony formation, as recently demonstrated using of an in vitro assay for quantification of stem cell/progenitor activity [29].

### 2.3. Lessons from SFK Inhibitors in AML

Over the years, SFK inhibitors have been developed. Several have entered clinical trials, and some have obtained FDA approval for cancer treatment. These inhibitors are small chemical compounds which target the ATP-binding pocket of SFKs. Many tyrosine kinase inhibitors targeting SFKs have been used on cell lines, AML primary specimens, and in mouse models transplanted with human AML cells. The major limitation of these inhibitors has been their lack of selectivity: that is, they also inhibit many other kinases, including KIT and FLT3 wild-type and oncoproteins [38], in addition to SFKs. While this was well known to be the case for some inhibitors such as dasatinib or bosutinib, other inhibitors (for example, PP2, used in a large number of studies) were formerly considered to be selective, but are now known to be highly promiscuous. Accordingly, some early data have required reevaluation in the light of present knowledge.

Several studies have shown that SFK inhibitors impair the leukemic phenotype of AML cells. The SFK inhibitor PP2 was shown to inhibit proliferation of AML cell lines [22], and to induce apoptosis in AML patient samples [28]. PP2 also prevented growth of a 32D-FLT3-ITD model when injected into mice [35]. PD180970 and SKI-606 inhibitors also blocked proliferation of AML cell lines [33]. Bosutinib has been shown to sensitize AML cell lines to ATRA-induced differentiation [39], and a similar result was observed using a combination of PP2 and ATRA [40]. Dasatinib, used either alone [34,41], or in combination with existing therapies, including ATRA [42], conventional chemotherapy [43] or navitoclax [44], impairs AML cell proliferation. Another SFK inhibitor, RK-20449, inhibited growth of CD34^+^CD38^−^ AML cells in vitro, and reduced leukemic burden in mice engrafted with primary human AML cells [45]. Interestingly, secondary transplants from mice treated with RK-20449 did not show engraftment of human cells, indicating that leukemia-initiating cells (LICs) were eradicated by the treatment. These observations on LIC targeting were also seen following dasatinib treatment [29]. Taken together, these studies, using a variety of SFK inhibitors, strongly suggest that targeting SFKs might constitute a reasonable therapeutic approach in AML, complementing established treatments.

Despite caveats, SFK inhibitors have proved to be very useful tools for targeting these kinases in AML. The demonstration of SFK involvement, however, has required the use of independent but complementary technical approaches (e.g., knock-out models, dominant-negative mutants or RNA interference). This will be illustrated below using the example of AML harboring the FLT3-ITD oncoprotein.

### 2.4. SFKs and FLT3-ITD in AML

Inhibition of FLT3-ITD expression or the use of selective FLT3 inhibitors has demonstrated that SFK activation is dependent on FLT3 in all AML models studied, including mouse models, AML cell lines and patient samples.

SFKs activate a wide range of proteins and signaling pathways. With regard to FLT3-ITD oncogenic signaling, the major contribution of SFKs has been highlighted in two essential pathways driving cell proliferation and cell survival. These pathways involve two key players: STAT5 and CDK6 (Figure 1).

STAT5 is a crucial effector of FLT3-ITD oncogenic signaling [46]. STAT5 is a signaling molecule and transcription factor, which shuttles from the cytoplasm to the nucleus, and regulates transcription of numerous genes involved in cell survival, including PIM1 and PIM2 in AML. STAT proteins are activated by an upstream tyrosine kinase, the candidates in FLT3-ITD-positive AML being JAK kinases, SFKs, or FLT3 itself. STAT proteins are clearly recognized as SFK substrates in other contexts [47]. An earlier study [48] had suggested that neither SFKs nor JAKs were involved in the activation of STAT5, implying that FLT3-ITD itself might be responsible for phosphorylation of STAT5 on tyrosine; however, that study relied on the use of non-AML cells. Subsequently, several groups have demonstrated that SFKs are essential requirements for STAT5 activation downstream of FLT3-ITD, using both inhibitor and RNA interference methods. The specificity of SFK inhibition was first shown in a FLT3-ITD transfected mouse cell line, and later by RNA interference targeting LYN [35]. Using SU6656, an inhibitor that—unlike many SFK inhibitors—does not affect FLT3-ITD, and specific RNA targeting, HCK was recently shown to be the STAT5 tyrosine kinase in the MV4-11 AML cell line [30]. Importantly, unlike FLT3-ITD, it has been established that FLT3-TKD mutations (D835) do not activate STAT5, probably because they fail to bind and activate SFKs [49].

CDK4 and CDK6 are cyclin-dependent kinases responsible for progression of the G1 phase of the cell cycle. Two independent functional experimental approaches led to the conclusion that CDK6 is essential for proliferation in AML with the FLT3-ITD mutation. The first of these studies was a functional siRNA screen of all human kinases [30], while the second was a drug screen carried out on AML cell lines [50]. Selective RNA interference and selective inhibitors demonstrated that this category of AML is strictly dependent on CDK6 for the G1/S transition, leading to the conclusion that CDK6, but not CDK4, is the therapeutic target. FLT3-ITD inactivation, using either kinase inhibitors or small interfering RNAs, showed that elevated expression of CDK6 is induced by the FLT3 oncoprotein. Furthermore, the pathway responsible for high CDK6 expression is dependent on SFKs, but independent of STAT5, MAP-Kinases, or PI3-Kinase pathways [30]. Inactivation of CDK6 by RNA interference or kinase inhibitors, or inhibition of SFKs, resulted in similar phenotypes, with inhibition of cell proliferation due to G1/S cell cycle arrest. In the MV4-11 AML cell line, HCK was shown to be the SFK responsible for CDK6 overexpression downstream of FLT3-ITD, since only HCK targeting recapitulated the phenotypes (CDK6 expression and cell cycle arrest). However, it is conceivable that other SFKs might play this exact same role in other AML subtypes.

Independently, inhibition of STAT5 or CDK6 may be considered as novel strategies for AML management. However, SFK inhibition has the advantage of targeting both pathways. Furthermore, several SFK inhibitors also hit essential hematopoietic receptors such as FLT3 or KIT, which might prove of additional interest in attempts to eradicate leukemic cells.

### 2.5. SFKs and Oncogenic KIT in AML

In hematopoiesis, following the binding of KIT ligand, stem cell factor (SCF), KIT dimerizes and initiates the subsequent activation of downstream signaling pathways, which have been extensively studied in different cell types. Four main pathways have been described: the PI3-Kinase/AKT, RAS-ERK, PLC-γ and SFK signal transduction pathways [51]. SCF binding on the first three Ig-like extracellular domains of KIT induces the subsequent phosphorylation of the di-tyrosine motif Y568-Y570 in the juxtamembrane region, thus initiating signal transduction, including the recruitment of SFKs via their SH2 domain [52,53]. Once activated, SFKs participate in the circuit of KIT downstream signaling pathways. Indeed, using SU6656, a selective SFK inhibitor that does not target KIT [54], SFKs have been shown to participate in the activation of the MAP-Kinase ERK via SHC phosphorylation [55] as previously speculated in studies using juxtamembrane KIT mutants [52]. SFKs are also required in the activation of the PI3-Kinase/AKT-dependent pathway [56]. Moreover, SFKs have been involved in KIT-mediated activation of RAC and JNK [57], as well as in the phosphorylation of adaptor/scaffold proteins such as GAB2 and LAT2 [58,59]. However, the role of SFKs has yet to be fully delineated downstream of KIT. Indeed, some studies performed on transfected cell models have not yet been confirmed in an endogenous context. In addition, some conclusions relied on the effects of promiscuous kinase inhibitors, or on KIT tyrosine mutants lacking SFK binding sites, but these binding sites also recruit various other signaling proteins.

Interestingly, it has been shown that oncogenic forms of KIT, such as KIT^D816V^, hijack the wild-type signaling circuit of KIT to promote their tumorigenic potential [60,61,62]. New pathways are also mobilised by oncogenic KIT mutants [63,64]. For instance, while the activation of the STAT transduction pathway by KIT^WT^ is highly transient [65,66,67,68,69], STATs are strongly and permanently activated downstream of KIT mutants, and their activation requires an intact juxtamembrane domain [61,63,70]. In addition, SFKs are constitutively activated downstream of KIT^D816V^, and are involved in its oncogenic activity [63].

Interestingly, two of the main KIT mutants identified in CBF-AML, KIT^D816V^ and KIT^N822K^ [71], generate different downstream signaling pathways, since KIT^D816V^ activates SFKs, whereas KIT^N822K^ activates ERK, a pathway that is downregulated by KIT^D816V^ (and KIT^WT^) [63,70]. Although KIT^D816V^ is associated with a poorer prognosis than KIT^N822K^ [72,73], the contribution of SFKs to this difference is unclear, as their inhibition by SU6656 affects the cellular proliferation of both mutants in liquid culture equally [70]. However, the colony formation assay, on semi-solid media, would be a more suitable technique for assessing the leukemic potential of KIT mutants.

The role of SFKs downstream of KIT in AML remains insufficiently understood. Since KIT is highly expressed in most AML cases, and is a recognized therapeutic target, it is of interest that some of the SFK inhibitors in clinical use, e.g., midostaurin, dasatinib, bosutinib, and ponatinib, also target KIT.

## 3. SFKs in Mastocytosis

Mastocytosis is a rare heterogeneous disorder associated with abnormal clonal expansion and accumulation of mast cells in various tissues. There are three major forms of mastocytosis as defined by the World Health Organization (WHO): mast cell sarcoma, cutaneous mastocytosis (CM), and systemic mastocytosis (SM) [74,75]. This last is subdivided into indolent systemic mastocytosis (ISM) and advanced systemic mastocytosis (AdvSM). CM and ISM are the most common subtypes, and have a favorable prognosis, while AdvSM has a poor prognosis, due to tissue infiltration by neoplastic mast cells leading to organ dysfunction (for review: [76]).

Mast cells are known to be essential for host defence responses, but also to play a role in allergy and anaphylaxis. They express a variety of receptors, including FcεRI and KIT, in order to respond to endogenous and exogenous stimuli [77]. The high affinity surface receptor for IgE, FcεRI, does not possess any intrinsic enzymatic activity, and thus relies on the recruitment of SFKs to trigger the earliest stages of activation following FcεRI aggregation. Transmembrane adaptor proteins become associated with FcεRI: the most studied are LAT (linker for activation of T cell) and NTAL (non-T cell activation linker) [78,79].

Mast cells are the only terminally differentiated hematopoietic-derived cells to express the stem and progenitor cell marker, the KIT receptor. The predominant genetic alteration identified in more than 80% of adult patients with mastocytosis is a somatic point mutation in the c-KIT gene, leading to an amino-acid change at position 816. This mutant, KIT^D816V^, induces a structural change in the catalytic domain, resulting in a permanent structural active state, and thereby the constitutive activation of the receptor tyrosine kinase [75,80].

### 3.1. SFKs and KIT Signaling

As described in Section 2.5, the KIT receptor recruits SFKs through the interaction of SFK SH2 domains at the juxtamembrane tyrosine docking sites. In mast cells, most studies of SFK responses to the KIT ligand (stem cell factor; SCF) have focused on LYN. By using lyn^−/−^ mouse models, the role of LYN in bone marrow-derived mast cell (BMMC) in vitro proliferation, degranulation (histamine release) and calcium mobilization has been studied by various groups. However, their findings were, and remain, controversial, with reports of LYN promoting, decreasing or having no effect on the same function and signaling transduction pathways [81,82,83,84,85,86,87,88,89,90,91,92], as summarized in Table 1. 

Numerous hypotheses, such as the use of different mouse model backgrounds, or the number of passages at which BMMCs were used [85], have been put forward to explain these discrepancies, underlining the need for additional studies. Finally, studies using a hck^−/−^ mouse model have identified HCK as an important factor for optimal mast cell proliferation, downstream of KIT^WT^ [90].

Signaling transduction pathways downstream of KIT^WT^ in mast cells are similar to those described previously in normal hematopoietic progenitor cells (see Section 2.5), apart from the addition of the STAT signaling pathway [93]. Indeed, SCF induces phosphorylation of STAT1, STAT3 and STAT5 in BMMCs [63,94], and the activation of this pathway, crucial for mast homeostasis [95], is dependent on JAK2. Interestingly, under other RTKs, such as EGFR, PDGFR or FLT3, SFKs are also involved in STAT activation [47,96] (see above for FLT3). While inhibition of JAK2 or JAK3 using RNA interference or various JAK inhibitors does affect the activation of STATs in neoplastic mast cells carrying KIT^D816V^, the SFK inhibitor SU6656 induces a downregulation of STAT1 and STAT3 phosphorylation, but not of STAT5 phosphorylation [63,97]. The significance of STAT1 and STAT3 tyrosine phosphorylation is unclear at this stage, because only STAT5 appears to be transcriptionally active in these cells [63,98,99].

Gleixner et al. reported the constitutive activation of LYN in advanced systemic mastocytosis (AdvSM); it was less frequently activated in indolent systemic mastocytosis (ISM) patient samples [10]. HCK was also found to be active in HMC-1, a cell line harbouring the KIT^D816V^ mutation, established from a patient with mast cell leukemia. Importantly, in the neoplastic mast cell lines, the activation of LYN was independent of the oncogenic KIT receptor [10]. The downregulation of LYN or HCK expression in HMC-1, using specific RNA interferences, induced apoptosis [10]. Interestingly, this contrasts with the impaired proliferation reported in lyn^−/−^ KIT^WT^ BMMCs, which was linked to cell cycle and not to cell death [83,85]. Accordingly, the function of SFKs may differ depending on the physiological context.

### 3.2. SFKs and FcεRI Signaling

Wild-type KIT receptor, by itself, is unable to induce degranulation or cytokine production, but can enhance these cellular responses, mediated by the high affinity surface receptor for IgE, FcεRI (Figure 2).

To date four SFKs have been functionally involved downstream of FcεRI: LYN, FYN, HCK and FGR (for review: [100]); they are critical to IgE-mediated mast cell activation, as they initiate FcεRI receptor intracellular signaling.

LYN was the first SFK found to be associated with FcεRI, and to be activated following receptor aggregation [101]. LYN is then required for the transphosphorylation of FcεRI ITAMs (immunoreceptor tyrosine-based activation motifs), for the activation of SYK, another protein kinase crucial for the phosphorylation of the adaptor LAT, and for the full assembly of the FcεRI signaling framework [102,103]. LYN also contributes to the feedback regulation of the FcεRI receptor, by activating several inhibitory proteins, including SHIP-1, and the adaptor proteins DOK-1 and CBP.

FYN is also expressed in BMMCs, and is activated following its interaction with engaged FcεRI, which then induces its interaction with GAB2, promoting GAB2 phosphorylation and subsequent activation of the PI3Kinase/AKT transduction pathway [84]. As an upstream regulator of these pathways, FYN is a positive mediator of the degranulation response to FcεRI engagement (fyn^−/−^ BMMCs) [84,87]. FYN is also responsible for STAT5 phosphorylation downstream of FcεRI [104]. STAT5 is required for mast cell development and survival: STAT5 deficient mice have mast cells at birth but not when adult. STAT5 is not essential for the early activation response (degranulation), but is required to induce cytokine release (the inflammatory secondary response). Therefore, like LYN, FYN is a central node for FcεRI-evoked responses.

HCK is expressed in BMMCs at lower levels than LYN and FYN [90], and does not precipitate with FcεRI [84,90]. However, hck^−/−^ BMMCs have revealed that HCK is a positive mediator of FcεRI-mediated degranulation, and is an important activator of both JNK and p38 MAP-Kinases and the PI3Kinase/AKT transduction pathways, and, surprisingly, of SYK [90].

Finally, the most recent SFK to have been linked to FcεRI is FGR [105]. FGR interacts with FcεRI, and is involved in the activation of all three classical MAP-Kinases and the PI3Kinase/AKT transduction pathways, as well as SYK phosphorylation [105]. FGR overexpression enhances degranulation of BMMCs, and is important for the modulation of FcεRI-dependent responses [105,106].

The assembly of FcεRI subunits then initiates complex SFK signaling transduction, activating LYN, FYN, HCK and FGR. Their individual impact is not redundant as they activate different downstream signaling transduction pathway. These four SFKs are nevertheless interconnected, since their activity is dependent on other members of the family, as demonstrated by the study of mouse knock-out mast cells: FYN kinase activity is upregulated in lyn^-/-^ BMMCs [86], and LYN activity is upregulated in hck^−/−^ BMMCs [90].

KIT and FcεRI receptor signaling orchestrate mast cells responses. The addition of KIT-ligand to human or murine mast cells potentiates FcεRI-mediated degranulation [88,107,108]. One route of convergence for KIT and FcεRI is the adaptor protein NTAL/LAT2, which cooperates with LAT in mast cell degranulation. Indeed, NTAL is directly phosphorylated by KIT [108], as well as by LYN/SYK following FcεRI aggregation [58]. Moreover, LYN activation by KIT promotes FcεRI-mediated degranulation, by lowering the threshold required for FcεRI aggregation required, and by increasing phosphorylation of BTK, a downstream tyrosine kinase effector of FcεRI signaling [88,109,110].

### 3.3. SFK Inhibitors in Mastocytosis

As stated above, the inhibition of KIT^D816V^ activity does not affect LYN or BTK constitutive activation in AdvSM patient samples, but their down-regulation by RNA interference decreases survival in KIT^D816V^-expressing cell lines [10]. This observation positioned SFKs as attractive targets for the treatment of systemic mastocytosis. However, selective inhibition of the activity of SFKs alone, using an inhibitor such as bosutinib, has been shown to be ineffective as a stand-alone agent; their use in combination therapy may prove to be more valuable [111].

Other drugs that target SFKs in addition to other protein tyrosine kinases have been developed [76,112]. Unfortunately, the majority of these inhibitors (e.g., dasatinib) have induced side effects and failed to achieve long-lasting remission. There are, however, two major exceptions to this: midostaurin and masitinib have both been identified as emerging treatment options by WHO [75]. Midostaurin (PKC412), initially characterized as an inhibitor of protein kinase C, inhibits the catalytic activity of KIT, FLT3, SYK, and SFKs [113]. Remarkably, an overall response rate of 60% was achieved in all mastocytosis subtypes [114,115]. Masitinib also inhibits the enzymatic activity of a variety of kinases, such as KIT, FLT3, SFKs and BTK [116]. While masitinib has no impact on KIT^D816V^ activity, Lortholary and colleagues [117] observed a significant improvement of symptoms in patients with severe symptomatic indolent systemic mastocytosis. Interestingly, masitinib appears to block mast cell differentiation mainly through its action on LYN activity [117].

## 4. Conclusions and Perspectives

SFKs are therapeutic targets of interest both in AML and mastocytosis. Yet, some challenges remain in order to fully decipher their role(s) in these contexts, and then to translate this knowledge to the clinic.

In AML, SFK activation is a common feature of most samples, despite the diversity of this disease. There is evidence that some SFK inhibitors also impair LIC activity, which is a promising property. Indeed, relapse and resistance to current therapies are still the main challenge in AML, as leukemic cells are extremely plastic and they inevitably escape single agent therapy. An important strategy is therefore the use of combination regimens. The prevalent and empirical approach is to combine SFK inhibitors with established chemotherapy or another existing targeted therapy in AML, and such clinical trials are currently ongoing. More innovative and rational treatments based on experimental evidence may shortly come up from dedicated functional screens of drug libraries or genetic tools targeting the whole genome.

Mastocytosis is due to an excess number of hyperactivated mast cells in various tissues. Indeed, mast cells release numerous mediators from internal granules, an event responsible for the symptoms. SFK-targeting drugs impair mast cell degranulation, as a result of FcεRI signaling blockade. Here, the combination of drugs hitting both the proliferation and survival functions controlled by mutant KIT receptor, and FcεRI signaling using SFKinhibitors, is also a promising objective of research.

## Figures and Tables

**Figure 1 cancers-12-01996-f001:**
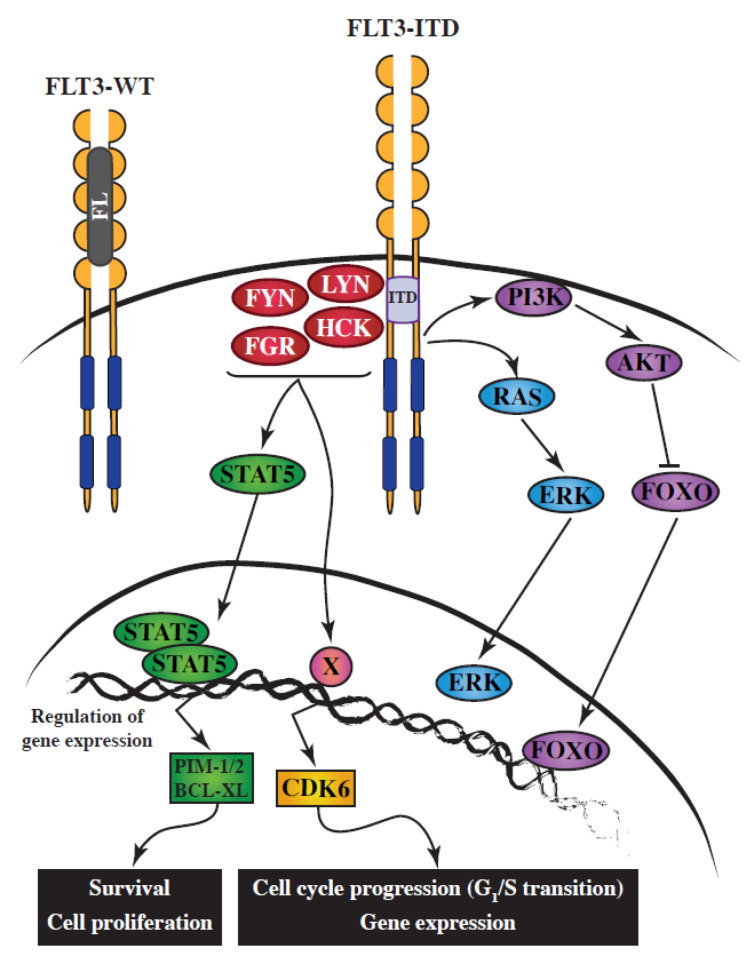
Simplified schematic representation of the implication of SRC-family kinases (SFKs) downstream of FLT3-ITD in acute myeloid leukemia (AML). FLT3 ligand (FL) binds the monomeric form of FLT3-WT, triggering its dimerization, which results in its autophosphorylation. Activated FLT3-WT then initiates multiple signaling transduction pathways. The aberrant activation of FLT3-ITD leads to the constitutive activation of SFKs, which are responsible for (1) STAT5 tyrosine phosphorylation, dimerization, and its subsequent translocation to the nucleus; and (2) the upregulation of CDK6 expression, in part through increased transcription. STAT5 and CDK6 are essential effectors of the FLT3-ITD oncoprotein.

**Figure 2 cancers-12-01996-f002:**
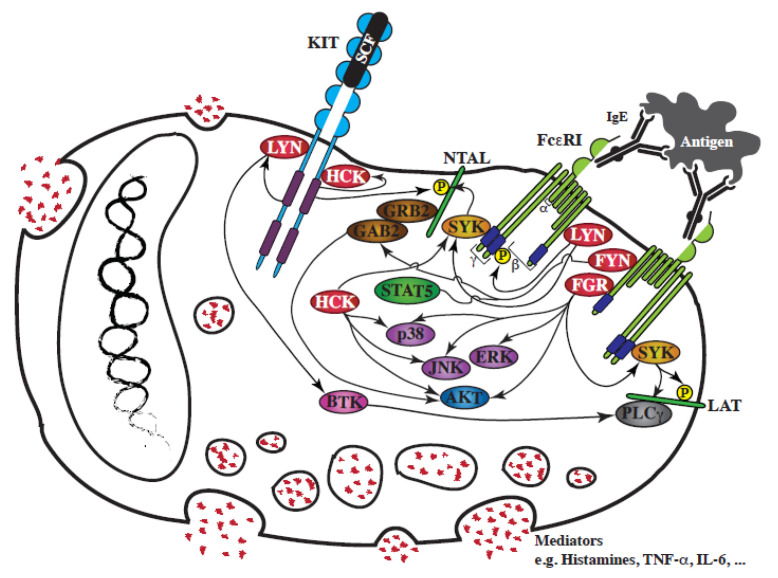
Schematic representation of the involvement of SFKs in FcεRI-mediated mast cell degranulation and in the interplay between KIT and FcεRI. Following SCF binding to the extracellular region of KIT, or FcεRI aggregation by antigen, a rapid activation of SFKs, LYN, FYN, HCK and FGR, is observed. FcεRI signaling regulates the degranulation of mast cell mediators stored in intracellular granules. KIT signaling synergizes with the FcεRI response by increasing SFK activation, which fuels several pathways activated downstream of FcεRI.

**Table 1 cancers-12-01996-t001:** Phenotypes observed in lyn^−/−^ mouse models compared to BMMC wild-type cells.

Ref.	In Response to SCF									In Response to Antigen								
	Cellular Functions			Signalling Pathways						Cellular Functions			Signalling Pathways					
	Proliferation (IL-3 and/or SCF)	Calcium Mobilisation	Chemotaxis	Phospho-AKT	Phospho-SHIP	Phospho-ERK1/2	Phospho-p38	Phospho-JNK	KIT Expression	Degranulation	Calcium Mobilisation	Cell Migration	Phospho-AKT	Phospho-SHIP	Phospho-ERK1/2	Phospho-p38	Phospho-JNK	FCεRI Expression
Nishizumi, 1997 [81]										=	↓				↓			=
Kawakami, 2000 [82]	=									delayed	↓				↑	=	↑	=
O’Laughlin-Bunner, 2001 [83]	↓		↓						=									
Parravicini, 2002 [84]										↑	↓		↑					=
Hernandez-Hansen, 2004 [85]	↑								=									=
Hernandez-Hansen, 2004 [86]										↑	delayed		↑	↓				
Odom, 2004 [87]										↑								↑
Iwaki, 2005 [88]		delayed		=		↓	=	=		↓	↓		↓		↑	=	↑	
Kitaura, 2005 [89]			↑									↓						
Hong, 2007 [90]										↑								
Poderycki, 2010 [91]	↑								=	↓		↓	↓	↓	↓	↓	↓	=
Ma, 2011 [92]	↑			↑	↓				↑									

(=) No effect observed; (↑) enhanced; (↓) decreased.

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
