# Peer review of "SRC-Family Kinases in Acute Myeloid Leukaemia and Mastocytosis"

_cancers, 2020, doi:10.3390/cancers12071996_

Round 1

Reviewer 1 Report

The review focuses on AML and mastocytosis, two hematological pathologies that are regulated by two related receptor tyrosine kinases: FLT3 and KIT. FLT3 is one of the most frequently mutated genes in AML, while KIT oncogenic mutations occur in 80-90% of mastocytosis. As the authors indicate, this review highlights the central roles of Src-related kinases (SFKs) in AML and mastocytosis, and their interconnection with FLT3 and KIT oncoproteins.

The collection of evidence is broad enough and well structured. The ideas are concise but  detailed. I find this is a useful review, as it addresses the connection between different elements and such connection is not easy to find in literature. I would only mention two minor aspects that, in my opinion, could be improved for the sake of clarity:

  • The molecular relationship between FLT3 receptor and SRKs should be showed earlier (with Figure 1). Current sub-section 2.4. (page 4), or at least lines 153-167 and figure 1, should appear earlier, I would even say that it should be first sub-section in section 2.
  • Table 1 (page 8) is quite informative, its design is however a bit confusing.

Author Response

  • The molecular relationship between FLT3 receptor and SRKs should be showed earlier (with Figure 1). Current sub-section 2.4. (page 4), or at least lines 153-167 and figure 1, should appear earlier, I would even say that it should be first sub-section in section 2.

Our response: We moved the beginning of sub-section 2.4 to the introduction of section 2 (lines 156-170; which we assume are the same as 153-167 pointed by the reviewer; we do not know why the numbering is different). This includes Figure 1.

  • Table 1 (page 8) is quite informative, its design is however a bit confusing.

Our response: We agree with reviewer 1 that the Table must be modified and optimized. We amended the table by transposing the axis, and indeed, the table is now much more readable.

Reviewer 2 Report

Voisset and colleagues present a comprehensive review on the role of SRC family kinases in AML and SM. The topic is of relevance for the field. The manuscript is a well written review of the current literature and describes the pathogenic role of SRC family kinases as downstream signaling molecules of mutant FLT3 and KIT. In vitro and in vivo studies and targeting concepts are discussed, and the review of the available literature is well balanced. I have no major criticisms.

Two additional aspect could be discussed in more detail:

  • Is expression or activation of SRC family kinases potentially a prognostic and/or predictive biomarker in AML or SM?
  • Is there any evidence for alternative activation of SRC family kinases (e.g. via other signaling cascades or via activating mutations) as mechanism of resistance to FLT3- or KIT-targeting drugs?

Author Response

  •     Is expression or activation of SRC family kinases potentially a prognostic and/or predictive biomarker in AML or SM?

Our response: Expression of some members of the SFK is a prognostic factors; this was indicated in lines 114-115. We changed that sentence to make the statement clearer.

Activation of SFKs occurs in most AML or SM samples, therefore the activation is not a prognostic or predictive marker.

  •     Is there any evidence for alternative activation of SRC family kinases (e.g. via other signaling cascades or via activating mutations) as mechanism of resistance to FLT3- or KIT-targeting drugs?

Our response: We are not aware of such a role of SFKs in resistance to FLT3- or KIT- targeting drugs. The main mechanisms of resistance reported so far are secondary mutations within the receptors themselves, the activation of other pathways, and the emergence of clones that do not have either FLT3 or KIT mutations.

Reviewer 3 Report

The aim of the project is to highlights the central roles of SRC-family kinase (SFK) in acute myeloid leukemia (AML) and mystocytosis and their interconnection with FLT3 and KIT oncoproteins.  

The review is not well-organized (especially in the beginning). The authors jump from subject to subject and it is rather confusing for the reader. 

I suggest to have a shorter and more general introduction, follow by a short chapter on SFK and their limitations. 

Then in the the SFKs in AML intro the authors introduce FLT3 but not KIT.   

I would change the title of chapter 3 I(page 7) n SFKs in Mastocytosis for consistency with chapter 2 (page 2). 

I suggest to review the manuscript for clarity and style.

I suggest to avoid using “promiscuous” in this review.  

Page 6, second paragraph I would delete “As mentioned in the introduction”. 

Page 7: second paragraph: I would delete “normal”. 

Author Response

  • The review is not well-organized (especially in the beginning). The authors jump from subject to subject and it is rather confusing for the reader. I suggest to have a shorter and more general introduction, follow by a short chapter on SFK and their limitations.

We are disconcerted by this comment; especially because the two other reviewers have pointed out that our manuscript was not only well structured (reviewer 1) but also well written (reviewer 2). Moreover, a scientific editor had already corrected our manuscript before submission.

  • Then in the the SFKs in AML intro the authors introduce FLT3 but not KIT.

We apologize for this oversight. We have now moved the introduction on KIT from section 2.5 to the beginning of chapter 2.

  • I would change the title of chapter 3 I(page 7) n SFKs in Mastocytosis for consistency with chapter 2 (page 2).

We agree and amended it.

  • I suggest to review the manuscript for clarity and style.

See above.

  • I suggest to avoid using “promiscuous” in this review.

We understand that “promiscuous” can be read as a funny word; however, it is the term used to describe unselective/undiscriminating inhibitors in the field, as testified by the two following -among many- papers:

-What Makes a Kinase Promiscuous for Inhibitors? Hanson SM et al, Cell Chem Biol, 2019. PMID: 30612951

- High-throughput Assays for Promiscuous Inhibitors. Feng BY et al, Nat Chem Biol, 2005. PMID: 16408018

  • Page 6, second paragraph I would delete “As mentioned in the introduction”.

We agree and deleted it.

  • Page 7: second paragraph: I would delete “normal”.

We agree and deleted it.